# Factorial Structure and Psychometric Analysis of the Persian Version of Perceived Competence Scale for Diabetes (PCSD-P)

**DOI:** 10.3390/bs9050050

**Published:** 2019-05-07

**Authors:** Habibeh Matin, Haidar Nadrian, Parvin Sarbakhsh, Abdolreza Shaghaghi

**Affiliations:** 1Health Education & Promotion Department, Faculty of Health, Tabriz University of Medical Sciences, Tabriz, P.C. 5166614711, Iran; habibehmatin@gmail.com; 2Social Determinants of Health Research Center, Tabriz University of Medical Sciences, Tabriz P.C. 5166614711, Iran; haidarnadrian@gmail.com; 3Biostatistics and Epidemiology Department, Faculty of Health, Tabriz University of Medical Sciences, Tabriz P.C. 5166614711, Iran; p.sarbakhsh@gmail.com; 4Medical Education Department, Education Development Centre (EDC), Tabriz University of Medical Sciences, Tabriz P.C. 5166614711, Iran

**Keywords:** validation, questionnaire design, self-perception, diabetes mellitus, self-care

## Abstract

As a basic psychological need, the level of perceived competence could expedite the achievement of diabetes self-management goals. Because of a lack of a specific data collection tool to measure the level of self-competence among Persian-speaking patients with diabetes, this study was conducted for (1) cross-cultural adaptation and (2) psychometric assessment of the Persian version of the Perceived Competence Scale for Diabetes (PCSD-P). Standard translation/back-translation procedure was carried out to prepare a preliminary draft of the PCSD-P. Content and face validities of the early draft were checked by an expert panel including 15 scholars in the field of health education and promotion as well as nursing education with experience of working and research on diabetes. The final drafted questionnaire was completed by 177 randomly selected patients with type 2 diabetes. On the basis of the collected data, the structural validity of the contrived version was appraised using exploratory and confirmatory factor analysis (EFA, CFA). Cronbach’s alpha and intraclass correlation (ICC) coefficients were used to check the scale’s reliability and internal consistency. The estimated measures of content validity index (CVI = 0.95) and content validity ratio (CVR = 0.8) were within the acceptable recommended range. The EFA analysis results demonstrated a single factor solution according to the items’ loadings for the corresponding component. The model fit indices, that is, root mean square error approximation (RMSEA = 0.000), comparative fit index (CFI = 1), Tucker–Lewis index (TLI = 1), incremental fit index (IFI = 1), normed fit index (NFI = 0.999), and relative fit index (RFI = 0.995), confirmed the consistency of the hypothesized one-factor solution. The values of the internal consistency and reliability coefficients were also in the vicinity of an acceptable range (α = 0.892, ICC = 0.886, P = 0.001). The study findings revealed good internal validity and applicability of the PCSD-P to measure the degree of self-competence among Persian-speaking type 2 diabetes patients to manage the chronic disease. Owing to unrepresentativeness of the study sample, future cross-cultural tests of PCSD-P are recommended on diverse and broader Persian-speaking populations.

## 1. Introduction

Perceived competence to perform disease management tasks and accomplish allied self-care expectations could play an important role in combating devastating complications of a lifelong persisting disorder such as diabetes. This subjective sense of capability could help patients with diabetes to better manage their disease, that is, to maintain the recommended level of blood glucose and prevent its related complications [1]. 

Successful management of disease and provision of required cares for patients with diabetes has been one of the major challenges for health systems in recent decades. Given the current number of recognized cases of type 2 diabetes (425 million) [2], it is a challenging task for many health care networks to consolidate limited resources for diabetes care and prevent the related complications [3].

Robust research evidence indicates reciprocal association between the perceived competence for self-management of type 2 diabetes and the measure of blood glucose level [1,4]. Competence is generally reflected on the patients’ nutritional behavior, physical activity pattern, stress control, and maintaining type 2 diabetes compatible life style [4,5].

In addition to the motivation and perceived support for individual autonomy [1], the perceived competence is closely related to the concept of self-efficacy [1] and is introduced as one of the important constructs of the self-determination theory (SDT) [6,7]. It was suggested that at least in the context of behavior modification, support of autonomy and motivation could independently lead to the adaption of health boosting behaviors and consequently better health profile [1]. 

Owing to a wide range of attributes that might affect control of blood glucose level in patients with type 2 diabetes such as severity of the disease, stressful life events, depression, social support, the patients’ socioeconomic status and baseline mental status, health care providers (HCPs) should consider the psychological needs of the patients by supporting their autonomy and respecting their viewpoints and dignity while simultaneously providing consistent information about the disease management skills. All these circumspections have favorable effects on the level of perceived competence among type 2 diabetes patients for self-management of their illness, acquisition of healthy behaviors, and maintenance of a decent lifestyle over the disease life course [1,4,6].

Different scales were invented to assess the perceived competence level, including measurement of Children’s Perceived Competence (CPCS) [8], perceived competence for patient-centered obesity counseling [9], physical activity [10,11], health related quality of life [12], health behavior and health-related quality of life in patients with cardiovascular disease [13], as well as measurement of competence level for learning medical contents [14]. The Perceived Competence Scale (PCS) is another tool for measurement of perceived competence based on the SDT and has four short parts. The PCS is a consistent instrument in different behavioral domains for the prediction of participants’ feelings or compliance with certain commitments [15]. 

The Perceived Competence Scale for Diabetes (PCSD), which was derived from PCS, had been developed by Williams et al. [16] and could be applicable for assessment of the type 2 diabetes patients’ self-competence in managing their disease and regulating their daily activities. The PCSD had been used in various studies on perceived competence of patients with type 2 diabetes and its impact on the disease control and management was dissertated [1,16,17,18,19,20,21]. The PCSD was also psychometrically tested in different languages [16,20,22]. As the PCSD was not validated before for use in the Persian language, and because of a growing number of type 2 diabetes cases in Iran [23,24,25,26] and other Persian speaking countries, for example Afghanistan [27] and Tajikistan [28], this study aimed to translate and assess psychometric properties of the PCSD-P for use in Persian-speaking patients with type 2 diabetes.

## 2. Materials and Methods

### 2.1. Study Objectives 

The main purpose of this study was to translate and psychometric analysis of the Persian version of the Perceived Competence Scale for Diabetes (PCSD-P). Face and content validity appraisal and reliability assessment of the scale were executed according to the standard procedures [29,30,31], and exploratory factor analysis (EFA) and confirmatory factor analysis (CFA) were performed to test the structural validity of the instrument. Internal consistency and reproducibility of the measure were assessed using the Cronbach α and test–retest intraclass correlation (ICC) coefficients. 

### 2.2. Study Sample

The study participants were 177 randomly selected registered patients with type 2 diabetes in the diabetes clinic of the Shahid Madani Hospital in the city of Khoy, northwest of Iran. The sample size was decided in order to comply with the recommended number of cases per item (at least five) to ensure accuracy of the factor analysis process [32,33]. The inclusion criteria included having an active profile in the Diabetes Clinic, over 30 years of age, and native Iranian nationality. The exclusion criteria were emigration to other provinces or countries during the study period, hospitalization due to severe disabling conditions, considerable limiting mental disorders such as Alzheimer’s disease or congenital mental retardation, or having a severe limiting disability such as quadriplegia or limiting cardiovascular disease. All these criteria were checked by the research team to guarantee the sampling precision.

### 2.3. Measurements

The original PCSD [16] is a four-item tool that could measure the self-perceived ability of type 2 diabetes patients in controlling and management of several aspects of their disease on a daily basis. The respondents rate their degree of agreement with each item of the questionnaire on a seven-point scale (from 1 representing not at all to 7 representing completely true). The lowest achievable total score of the scale is 4, which suggests the lowest perceived level of competence, and the highest average score is 28, which indicates a higher perceived self-care competence [17,22,34].

The standard translation/back-translation procedure was applied [35,36] to translate the original English version of the PCSD into Persian (PCDS-P). Two fluent translators at the first stage translated the English PCSD into Persian and two other proficient translators back translated the Persian version into English. At the latest stage of the process, the prepared back-translated English version of the PCSD was compared with the original version, minor corrections were made, and the final draft was approved by the research team. 

Content validity of the PCSD-P was appraised quantitatively and qualitatively by sending the final version of the PCSD-P to a group of experts including 15 specialists in the field of health education and promotion and also nursing staff in the diabetes clinic. On the basis of the experts’ feedbacks about relevancy, as well as the clearness and lucidity of the wordings, Lawshe’s item-level content validity ratio (CVR) and the instrument level content validity index (CVI) were calculated; all of which were within the acceptable range (CVI ≥ 0.8 and CVR = 0.49) [29,30].

To assess internal consistency and reliability of the PCSD-P, Cronbach’s alpha and ICC were also measured (ICC was calculated after completing the PCSD-P by 20 patients twice in 20 days interval).

**Data collection:** The PCSD-P was completed by the trained interviewers for all respondents in a face-to-face interview in a non-directive manner and in one of the clinic’s separate and private rooms. The average scale’s completion time was 10–15 minutes. During the interview session, the participants were also questioned about their age, sex, marital status, occupation, level of education, permanent living place, nationality, and income.

### 2.4. Procedure and Ethical Considerations

Ethical approval for this study was obtained from the Medical Ethics Board of Trustees in the Tabriz University of Medical Sciences (approval number: IR.TBZMED.REC.1396.192). All of the study participants were provided information about the study objectives, their right to withdraw at any stage without jeopardizing their routine diabetes care or obligation to give reason, and also about confidentiality of the data at the beginning of interview sessions. The written informed consent was obtained from all the attendants or their guardians. 

### 2.5. Statistical Analysis

The study participants’ PCSD-P scores distribution was checked for skewness (standard value between −1 and +1) and kurtosis (standard value of +1.96 to −1.96) and the mean and standard deviations of the scores were estimated [37]. The PCSD-P scores were also examined for ceiling and floor effects, that is, to check whether more than 15% of the study participants achieved the highest and lowest possible scores, respectively [32].

The scale’s estimated Cronbach’s alpha reliability index was deemed satisfactory with a value above the threshold level of 0.7, and the ICC coefficient as an intuitive measure of the scale’s test–retest consistency was approved with its value above 0.61 [32]. 

To check construct validity of the PCSD-P, exploratory and confirmatory factor analyses (EFA and CFA) were performed. Calculation of the Keyser–Meyer–Olkin (KMO) index of sampling adequacy and Bartlett’s Test of Sphericity to confirm existence of patterned correlation among the scale’s items were carried out in the EFA string using the extraction method of principal axis factoring (PAF) and varimax rotation. Eigenvalues greater than 1 and factor loadings above 0.3 at this stage were considered significant and used for the factor(s) analysis [32]. In the confirmatory factor analysis, the considered acceptable ranges were 0.5–0.8 for the root mean square error approximation (RMSEA); 0.90–0.95 for the comparative fit index (CFI); χ^2^ < 3 for the value of chi square test; and values above 0.95 for the indices of Bentler-Bonett Normed Fit (NFI), Relative Fit (RFI), Incremental Fit Index (IFI), and Tucker–Lewis Index (TLI) [38,39]. To improve the item-level goodness of fit values and with the modification indices (MI) greater than 7, residual covariance was added to the model [40,41]. The Statistical Package for the Social Sciences (SPSS) (IBM SPSS, Version 20, IBM Corporation, New York, United States) and its added module Analysis of a Moment Structures (AMOS) (IBM SPSS Amos Version 22, Amos Development Corporation, Chicago, United States) were used for statistical analysis. 

## 3. Results

### 3.1. Sample Characteristics 

Skewness test of the study data revealed a symmetric pattern in the range of a normal distribution (−0.77). The mean total PCSD-P score of the study attendants was 20.79 with the standard deviation of ±6.46. Among the study participants, 3 individuals (1.7%) had the lowest score (4) and 40 participants (22.6%) achieved the highest score (28). 

The mean age and standard deviation of the study sample was 57.49 ± 11.57 years of which 64.4% were female and 85.3% were married. Other attributes of the participants are summarized in Table 1.

### 3.2. Content and Face Validity

The calculated CVI and CVR based on the experts’ feedbacks verified the content and face validity of the PCSD-P. The values of CVR for three questions were higher than 0.8 and above 0.6 for one of the questions. The total CVI score of the scale was 0.95, which is within the acceptable range [30]. 

### 3.3. Construct Validity

The EFA preliminary results based on the KMO measure of sampling adequacy (0.842) and value of the χ^2^ (748.414, df = 6, P = 0.000) verified the suitability of performing the analysis on the study data. A single factor four-item model fit was identified with all the factor loadings above 0.92, as indicated in Table 2. The pinpointed factor explained 87.470% of the total variance within the study data. The model fit was consistent with the empirically derived model in other psychometric studies of the PCSD [16,22,42]. The conducted CFA also confirmed the structure of the extracted single factor model in the EFA phase (RMSEA = 0.000, CFI = 1, TLI = 1, IFI = 1, NFI = 0.999 RFI = 0.995) (Figure 1). 

### 3.4. Reliability

The estimated Cronbach’s alpha measure of reliability (0.892) and the test–retest (ICC) measure of the scale’s consistency and stability over time (0.865, P = 0.001) were in the acceptable ranges.

## 4. Discussion

The main purpose of this study was psychometric analysis of the Persian version of PCSD (PCSD-P) to be used in research and practice settings for determining the degree of perceived competence for self-care among type 2 diabetes patients. The results revealed a good and acceptable psychometric property for application in Persian speaking patients suffering from type 2 diabetes. On the basis of the results, which were almost consistent with other studies on the psychometric assessment of the PCSD in different languages [16,20,22], this scale can be used as a proper tool for pre-assessment of the patients with type 2 diabetes in clinical or research settings, or post-intervention impact assessments of empowerment intervention targeting type 2 diabetes patients. As with the other translated versions, the PCSD-P demonstrated a robust psychometric performance for use in diabetes related topics and assessment of self-efficacy and sufficiency in overcoming barriers [1,17,18,21].

**Content validity:** The estimated reliability and internal consistency index of Cronbach’s alpha for PCSD-P in this study were almost in the range of calculated measure in other psychometric studies of the PCSD (α > 0.8) [16,20,22,42]. The approved unidimensionality of the instrument in this study was identical to the results obtained in other studies [16,22,42].

## 5. Limitation

The cross-sectional design of this study prevented the study team from checking whether the degree of perceived competence for self-care among the studied type 2 diabetes patients could lead to a long-term better self-care behavioral profile. The mere reliance on cognitive perception of the patients for self-care and avoiding questions about actual disease oriented self-management behavioral pattern, which could lead in turn to a germane glycemic control algorithm, are the most important drawbacks of the instrument.

At the time of this study implementation, the researchers were not aware of the degree of participants’ success in their disease management and this could be a potential confounder that might pose an effect on the study attendants’ responses. Patients with a better health profile could have a comparable answer pattern when compared with those who had a worse health profile. This is somehow related to the overall mentality of the patients in the time of responding to questions rather than their actual perceived self-competence.

Despite all these limitations, the consistency of the findings with the results of other studies suggest the applicability of the brief scale as an efficient measure to examine the perceived level of competence for self-care in patients with type 2 diabetes and possibly relate the results to patients’ success in better control of glucose level, which is crucial for prevention of the disease’s costly and non-reversible complications. 

## 6. Conclusions

The current study results provide initial support for the use of the PCSD-P in Persian-speaking type 2 diabetes patients by clinicians and researchers. The findings illustrated the applicability of the PCDS-P to assess patients’ perceived competence for self-care and approved the previously reported psychometric properties of the scale in other populations. The results also offered an insight into measurement of a generally neglected attribute in care provision for sufferers of one of the millennium chronic and devastating diseases. The scale could also be applicable as a generic adaptable template for use with other chronic medical conditions. Further research is recommended for cross-cultural validation of the instrument to its application in wider and international scope for comparison purposes. Future studies could address the sensitivity and predictive validity of the scale in measurement of perceived competence changes over time and self-care outcome when self-care competency improvement interventions are being taken for overall health promotion purposes. 

## Figures and Tables

**Figure 1 behavsci-09-00050-f001:**
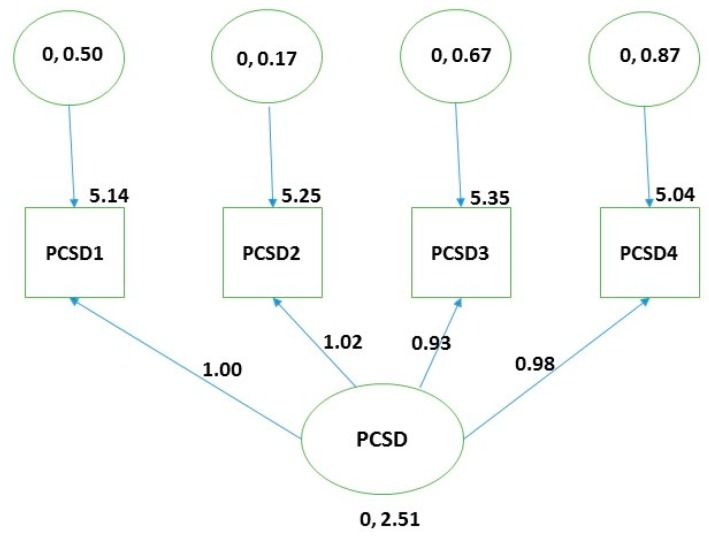
Visual representation of the items’ loadings in one component model obtained from confirmatory factor analysis (n = 177) in the psychometric appraisal of the Persian Perceived Competence Scale for Diabetes (PCSD-P).

**Table 1 behavsci-09-00050-t001:** Socio-demographic characteristics of the study participants to assess psychometric properties of the Perceived Competence Scale for Diabetes in Persian (PCSD-P).

Characteristics		Frequency	Percent
Gender	Male	63	35.6
	Female	114	64.4
Marital Status	Single	2	1.1
	Married	151	85.3
	Widowed	22	12.4
	Divorced	2	1.1
Occupation	Employees	10	5.6
	Retired	26	14.7
	Self-employed	23	13
	Housewife	104	58
	Unemployed	3	1.7
	Farmers/Stockbreeder	11	6.2
Education	Illiterate	63	35.6
	Primary education	44	24
	Secondary education	23	13
	High school level	30	16.9
	Post-graduate degree	17	9.6
Place of residence	Urban	145	81.9
	Rural	32	18.1
Income level (RLs: The Iranian national currency)	<15 million	106	59.9
	≥15 million	69	39
	Without any income	2	1.1

**Table 2 behavsci-09-00050-t002:** Eigenvalues and fit indices in the Psychometric appraisal of the Perceived Competence Scale for Diabetes in Persian (PCSD-P).

Item	EFA Loadings
PCSD1	0.927
PCSD2	0.953
PCSD3	0.934
PCSD4	0.927

EFA: Exploratory Factor Analyses. Extraction method: principal axis factoring. Rotation method: varimax, values higher than 0.3 were considered for inclusion.

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
