# Peer review of "Factorial Structure and Psychometric Analysis of the Persian Version of Perceived Competence Scale for Diabetes (PCSD-P)"

_behavsci, 2019, doi:10.3390/bs9050050_

Reviewer 1 Report

Before anything else,English language should be corrected. The text is full of grammatical mistakes, wrong punctuations while some sentences do not make sense (for instance, lines 83, 87, 90, 202, 217 and many more).

Author Response

 Point 1: Before anything else,English language should be corrected. The text is full of grammatical mistakes, wrong punctuations while some sentences do not make sense (for instance, lines 83, 87, 90, 202, 217 and many more).

 Response 1: Many tanks respected reviewer for this comments;

This manuscript was edited both by the Translation Institute and research team, if the respected reviewer still believe English language editing is necessary, we will do it.

Reviewer 2 Report

In the research titled Factorial structure and psychometric analysis of the Persian version of perceived competence scale for diabetes (PCSD-P), the authors aimed to translate the PCSD to Persian language and to assess its psychometric properties for use in Persian speaking type 2 diabetes patients. The study study was written with adequate detail and clarity. The authors have used valid methods for the assessment of instrument validity and consistency.

I would suggest that the authors specify the objective of the study in the abstract clearly following the brief background. 

In addition, the authors should first define abbreviations such as ICC at first use and consistently use the abbreviation after that. For all other abbreviations, I suggest to follow the same rule.

The name of the fourth column of table 1 should be percent instead of range.

Figure 1 ( path analysis) is barely visible in the pdf document. 

In the discussion, I suggest to include public health implication of the research.

Author Response

Many tanks respected reviewer for this comments

Point 1: I would suggest that the authors specify the objective of the study in the abstract clearly following the brief background. 

 Response 1:The objective of the study as mentioned in the text was (1) cross-cultural adaptation and (2) psychometric assessment of the Persian version of the Perceived Competence Scale for Diabetes (PCSD-P); that specified in the abstract by the numbers.

 Point 2: In addition, the authors should first define abbreviations such as ICC at first use and consistently use the abbreviation after that. For all other abbreviations, I suggest to follow the same rule.

 Response 2:According this comment, all abbreviations were defined at first.

  Point 3: The name of the fourth column of table 1 should be percent instead of range.

 Response 3:In the fourth column of table 1, Percent has been replaced instead of range.

 Point 4: Figure 1 ( path analysis) is barely visible in the pdf document. 

 Response 4: Path analysis was drawn in PowerPoint at first, then it turned into a JPEG format. Slide of this figure is upload.

Point 5: In the discussion, I suggest to include public health implication of the research.

Response 5: Public health implication was explained in line 10-13 of discussion.

Round  2

Reviewer 1 Report

the paper shows much improvement